# Evaluation of Waste Express Bag as a Novel Bitumen Modifier

**Yuming Lin [1], Chichun Hu [1,\*], Sanjeev Adhikari [2], Chuanhai Wu [3] and Miao Yu [4]**

1   School of Civil Engineering and Transportation, South China University of Technology, Guangzhou 510641, China; linyuming@outlook.com
2   Construction Engineering and Management Technology-Department of Engineering Technology, Purdue University, West Lafayette, IN 47907, USA; sanadhik@iu.edu
3   Guangdong Hualu Transportation Research Institute, Guangzhou 510420, China; hualuwuchunhai@outlook.com
4   Michigan Technological University, Houghton, MI 49921, USA; myu2@mtu.edu
\*   Correspondence: cthu@scut.edu.cn; Tel.: +86-138-2604-0612

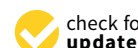

**Featured Application: This work provides reference for selecting appropriate approaches to analyze the characteristics of Waste Express Bags modified bitumen.**

**Abstract:** With the rapid development of China's e-commerce and logistics industry, a large number of waste express bags (WEBs) have been produced, which are difficult to recycle. The existing methods of waste express bag disposal often cause severe environmental pollution. It was discovered in this paper that the use of WEBs to modify bitumen could be an environmentally friendly way to recycle WEBs. This study aimed to investigate the feasibility of using WEBs to modify bitumen and promote the performance of WEB-modified bitumen. In order to verify this assumption, a series of basic or rheological experiments were conducted on different dosages of WEB-modified bitumen. The test results and phenomenon showed that the WEBs could be used to modify the binder homogeneously by using a high-speed shearing machine. The basic experiments showed that the WEBs could decrease the penetration and ductility while increasing the softening point and the rotational viscosity. Additionally, the rheological experimental data indicated that the high temperature performance was improved while the low temperature performance stayed the same. The Fourier-transform infrared spectroscopy (FTIR) results demonstrated that modification of the WEB was a physical modification without chemical reaction and the main component of the modifier was polyethylene. The fluorescence microscope (FM) data revealed the micro-structures of different dosages of WEB-modified bitumen. In conclusion, WEB can be a feasible binder modifier.

**Keywords:** waste express bag; multiple stress creep recovery test (MSCR); bitumen modifier; performance evaluation

## 1. Introduction

With the rapid development of China's e-commerce economy, online shopping has become an indispensable part of Chinese life. Inevitably, a large amount of express waste will be produced. Express waste can be divided into three categories: express bags, cartons and tape. Among them, cartons can be recycled and reused very efficiently. About 50% of cartons can be reused in the US, and the number in Europe is 80% [1]. Express bags are sealed with sealing strips for transportation purposes. However, as the express bag is sealed, people have to tear open the bag in order to take out the goods. Therefore, waste express bags (WEBs) cannot be reused like normal plastic bags. In China,

around 8.68 billion express bags were used in 2015 and this number has increased to 14.7 billion in 2016 [2,3]. The huge number of WEBs can only be disposed of by landfill and incineration, which will cause serious environmental problems.

Bitumen, as the most common paving materials in the world, can not only satisfy the basic function of road construction, but can also be used to deal with domestic garbage and industrial refuse. For example, some studies have used waste tire rubber [4] to modify bitumen, while some have used waste bottle plastic [5] or lignin [6]. Even waste cooking oil and waste glass can be used as modifiers [7,8]. This garbage can also improve the performance of bitumen, for instance, the high temperature performance and fatigue resistance of the rubber modified bitumen is better than that of normal virgin bitumen [9]. From this perspective, using the WEBs as the modifier should also have some positive effect on the virgin bitumen.

As a lightweight plastic bag, the main components of WEB are low-density polyethylene (LDPE), hot melt-adhesive, titanium dioxide (($TiO_2$), and calcium carbonate ($CaCO_3$)). Among them, the quality of PE accounts for about 85%, while the hot melt adhesive accounts for about 10%. The calcium carbonate and titanium dioxide, which are considered stain materials, account for about 5%. The hot melt adhesive mentioned here is a kind of pressure sensitive hot melt-adhesive, which is used as the sealing strip, the main component of which is polyolefin (PO). LDPE, which has a density of 0.915 $g/cm^3$ and a melting point of around 120–125 °C has been used as a modifier in previous research. Tan et.al found that mixed LDPE with the SMA mixtures could enhance the performance of SMA at both high-temperature and low temperature environments. It was also found that LDPE improved the stiffness and modulus of rupture values of the asphalt mixtures at low temperature [10]. Fang and Hu tried to compare the EPS modified bitumen and PE modified bitumen [11] and used FTIR to indicate that there was no obvious change with the main functional groups in the waste PE modified bitumen. The result confirmed the modification mechanism of waste PE was physical process. Fang and Li tried to mix waste PE and rubber to modify the asphalt matrix and study the properties of the modified bitumen [12]. In this study, the network structure of WPE formed by cross-linking action may be the reason for the improvement of its high-temperature performance. The paper also showed the combined modification of bitumen by WPE and the waste rubber powder had a better performance. Polyolefin, as the main component of the hot melt adhesive with the density of 0.91 $g/cm^3$ and melting point around 135 °C, has also been used to modify bitumen. Guan used polyolefin to modify the bitumen and applied it in an ultrathin friction course [13]. Hesp studied the stabilization mechanisms in polyolefin-bitumen emulsions [14].

The WEB stain, with the main component as $TiO_2$, can also be used to modify the binder. The density of $TiO_2$ is 4.23 $g/cm^3$ while its melting point is over 1800 °C. It has been demonstrated that $TiO_2$ can absorb the nitrogen oxide in the environment by the catalysis of ultraviolet light. For example, Liu et.al conducted a series of tests on how to maximize the degradation of the nitrogen oxide by using $TiO_2$ as the modifier [15]. Zhang tried to evaluate the high temperature and low temperature properties of nano-$TiO_2$ SBS modified bitumen [16]. The density of calcium carbonate is 2.93 $g/cm^3$ and the melting point is 1339 °C. Some previous studies have indicated the calcium carbonate can also be used as a modifier [17]. Hao found that nanocalcium carbonate could be used as a modifier to improve both the high temperature performance and water stability of asphalt concrete [18].

Although some of these materials were individually used to modify the bitumen in previous studies, the effects of their combination in a specific ratio are still unknown. It is important to consider that, as a form of garbage, many of their properties are different to those used by other scholars. All of the above materials have had positive effects on the bitumen binder. Therefore, their combination can also be anticipated to have good effects.

The purpose of this study was to evaluate the feasibility of using WEB as a novel bitumen modifier for hot mix asphalt. In this paper, basic experiments such as penetration, softening point, and ductility tests were conducted to evaluate the physical properties of different dosages of WEB-modified bitumen. The rheological properties such as rutting factor and viscosity were also studied in this paper.

Furthermore, this paper proposes an environmentally friendly and effective way to deal with WEB sand apply them to improve the performance of the bitumen binder.

## 2. Materials and Methods

### 2.1. Preparation of Materials

This study investigated the performance of the WEB-modified bitumen at different dosage levels (0% to 10%) when mixed into virgin bitumen. The general information of the virgin bitumen binder used in this experiment is shown in Table 1.

**Table 1.** Basic information of pen 60/70 bitumen.

| Test Item | Result | Specification |
|---|---|---|
| Penetration (25 °C), 0.1 mm | 63.4 | ASTM D5 |
| Penetration Index | −1.35 | ASTM C1125-89 |
| Softening point (R&B), °C | 49.0 | ASTM D36 |
| Ductility (15 °C), cm | >150 | ASTM D113 |
| Ductility (10 °C), cm | 31.5 | ASTM D113 |
| Kinematic Viscosity (60 °C), Pa *s | 208 | ASTM D445 |
| Wax content, % | <2.1 | T-0615 |
| Solubility in Trichloroethylene | >99.5 | ASTM D2042 |
| Flash point, °C | 335 | AASHTO T 79 |

The penetration grade of the bitumen binder was 60/70 (pen 60/70). The WEB used in this experiment was produced by the Shunfeng Company. The WEBs were colored white on the outside and black inside (Figure 1a). Figure 1b shows how the WEBs were ground using a grinding machine. The aperture of the filter was 2 mm. The WEB diameter after grinding was less than 2 mm (Figure 1b,c). The WEBs were then heated in an oven at 135 °C for 5 min and then mixed with the pen 60/70 bitumen binder. The WEB and virgin bitumen binder mixture was stirred homogeneously by a high-speed shearing machine (13,000 RPM) at a temperature of 180 °C ± 5 °C for 60 min. Figure 1d shows the modification of the WEB and virgin bitumen binder. The test results of previous research revealed that WEBs over 10% had poor workability and storage stability, which means that if the dosages of the WEB additives exceeded 10%, then the WEB-modified bitumen cannot meet the construction requirements [19]. The dosages of WEB were set as 0%, 2%, 4%, 6%, 8% and 10% by the mass of modified bitumen.

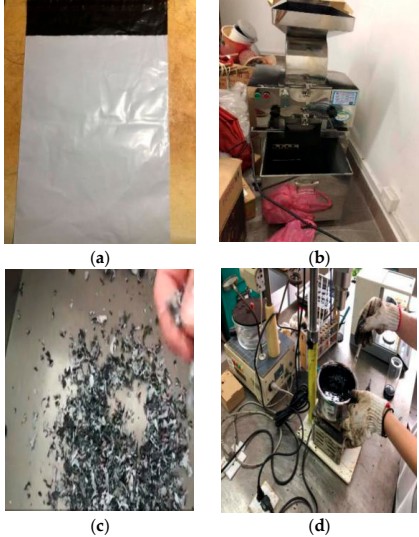

**Figure 1.** Preparation of the wasted express bag (WEB)-modified bitumen. (**a**) wasted express bags; (**b**) grinding machine; (**c**) wasted express bags (after grinding); (**d**) modification process.

## 2.2. Experimental Plan

The detailed experimental plan of this research is listed in Table 2. The basic performance of the WEB-modified binder was analyzed by three indicators: softening point, penetration, and ductility at 10 °C. The chemical mixing modification mechanism of the WEB-modified bitumen was analyzed by Fourier transform infrared spectroscopy (FTIR) and fluorescence microscope (FM) observations. In order to obtain more reliable results from the experiments, several test samples were prepared.

**Table 2.** Experimental plan of the WEB-modified binder.

| Performance | Tests | Aging Level | Specification |
| --- | --- | --- | --- |
| Physical properties | Penetration<br>Softening point<br>Ductility | unaged<br>unaged<br>unaged | ASTM D5<br>ASTM D36<br>ASTM D113 |
| Workability | RV | unaged | AASHTO T316 |
| High-temperature performance | Temperature sweep<br>MSCR | Unaged, RTFOT-aged<br>RTFOT-aged | AASHTO M320<br>AASHTO MP19 |
| Low-temperature performance | BBR | RTFOT-aged,<br>PAV-aged | AASHTO T313 |
| Modify mechanism and Morphology | FTIR<br>FM | unaged<br>unaged | N/A<br>N/A |

The high-temperature performance of WEB-modified bitumen was evaluated by the Superpave rutting factor (SRF) test and the multiple stress creep recovery test (MSCR). In both of these tests, a 1-mm gap and 25=mm-diameter plates were used. The SRF test was conducted at a temperature of 58 °C and the frequency was set at 10 rad/s. The amplitude of the test was 15%. During the SRF test, the temperature was changed at an interval of 6 °C (whether it was increasing or decreasing) until the rutting factors failed when the value was less than or equal to 1.0 kPa for un-aged bitumen or 2.2 kPa for aged bitumen (short term aging). The MSCR test was conducted at a temperature of 64 °C with two stress levels (0.1 kPa and 3.2 kPa) using RTFO aged binder. Three replicates were prepared and tested. During the RTFO test, 10 cycles of creep and recovery were conducted at each stress level that included one second for creep stress and nine seconds for recovery. The 0.1 kPa stress level was simulated for light traffic and the 3.2 kPa stress level was simulated for heavy traffic stress. The non-recoverable creep compliance ($J_{nr}$) was conducted at both stress levels, and the stress sensitivity parameter ($J_{nr-diff}$) and the average percent recovery (R) was calculated by the MSCR test.

The low-temperature performance of bitumen was evaluated by the bending beam rheometer test (BBR). The BBR test was conducted at temperatures of −6 °C and −12 °C, which follows AASHTO T313. During the BBR test, the temperature was changed at an interval of −6 °C (whether it was increasing or decreasing) until the result reached the standard (creep stiffness ≤ 300 MPa and m-value ≥ 0.3).

The workability of the bitumen was analyzed by the Brookfield rotational viscosity. The specification requires that the rotational viscosity does not exceed 3 pa.s. The test temperature was 135 °C, which is required by the specification. The experimental error was mainly due to the mechanical error. Therefore, the data were recorded after the value of the rotational viscosity tended to stabilize (the change did not exceed 30 mpa.s within one minute). The data were recorded once a minute in triplicate and the average value was selected.

## 3. Discussion

### 3.1. Three Empirical Indicators

The results of the penetration (25 °C), ductility (10 °C), and softening point tests are shown in Figure 2. Like most types of polymer modified bitumen, the softening point of the WEB-modified bitumen was increased while the penetration and ductility were decreased.

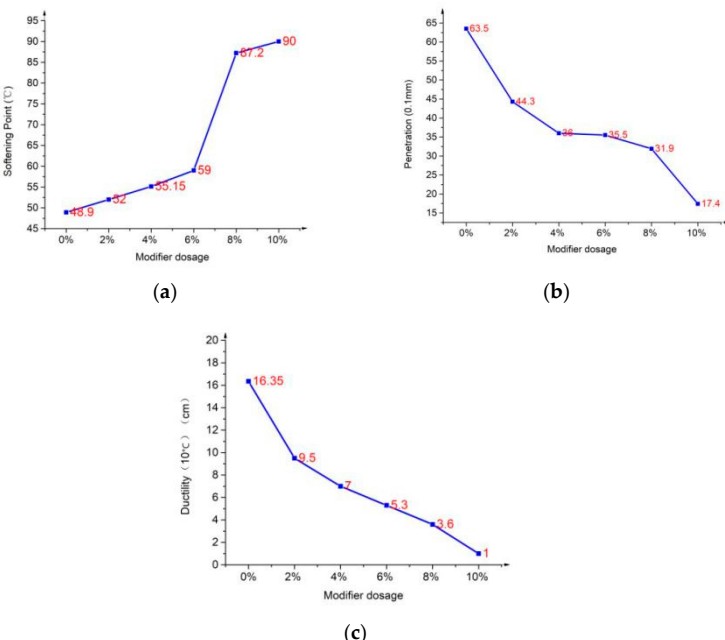

**Figure 2.** The three empirical characteristics: (**a**) softening point; (**b**) penetration; (**c**) ductility.

Compared with the virgin bitumen, the softening point of WEB-modified bitumen increased by about 3 °C after mixing the 2% WEB modifier in the initial stage. However, when the dosage of WEB modifier was 8%, the softening point was increased significantly and reached 90 °C. The result of the 10% WEB-modified bitumen was also over 90 °C. There are two possible reasons of this rapid increase: (A) When the modifier content reaches a certain level (>8%), the polyethylene (PE) components of the WEB cross-link within the matrix bitumen to form a spatial network structure, which effectively improves the high temperature performance of the bitumen; and (B) The content of calcium bicarbonate (CaCO$_3$) in waste express bags reaches a certain level where the modified bitumen becomes a modified bitumen mastic, thereby increasing its high temperature performance.

The penetration of the WEB-modified bitumen was less than 50, which was much smaller when compared to the pen 60/70. This phenomenon corresponded to the increase of the softening point. From a macro perspective, the bitumen became harder, which means it can resist more shear damage.

The ductility of the WEB-modified bitumen was also much lower when compared to the virgin bitumen, which indicates that the WEB-modified bitumen has poor tension resistance in a low temperature environment. Since previous research has shown that PE modified bitumen was simple physical blending, this phenomenon may relate to the WEB modifier. Qin had demonstrated that the tensile yield strength of PE will decrease with the increase of temperature in a low temperature environment (−30 °C, −10 °C) [20]. Therefore, with an increase in the WEB modifier, the tension resistance of WEB-modified bitumen will decrease.

### 3.2. High Temperature Performance

High temperature performance was evaluated by both the Superpave rutting factor test and the MSCR test. The main test data of the Superpave rutting factor test include the rutting factor, storage modulus, loss modulus and phase angle, which are shown in Figure 3. It can be seen that the rutting factor of WEB-modified bitumen was increased with an increase in the modifier dosage at the same temperature level. Similar to the softening point, the improvement of the rutting factor was stable in the range of 0–6%, however, when the dosage of the WEB modifier reached 8%, the rutting factor had a significant increase. This phenomenon is very interesting, which indicates that the high temperature performance of WEB-modified bitumen will be greatly improved when the dosage of the WEB modifier reaches a certain degree.

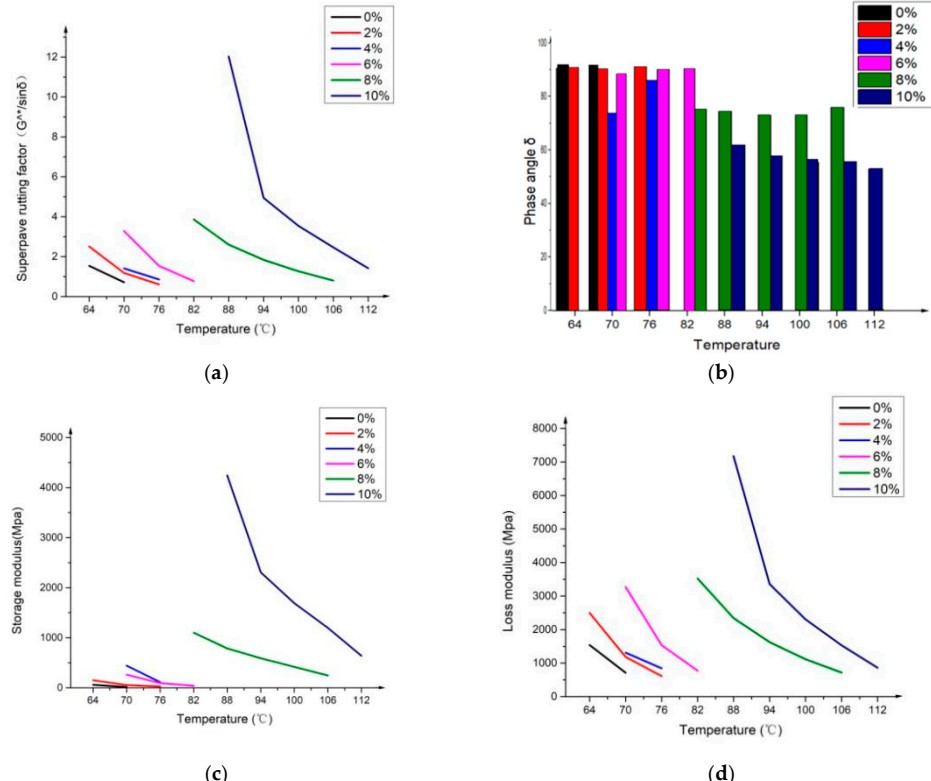

**Figure 3.** Results of the Superpave rutting factor test: (**a**) rutting factors; (**b**) phase angle; (**c**) storage modulus; and (**d**) loss modulus.

In this research, the test temperatures of different dosages of WEB-modified bitumen were different. There were two reasons for this design. First, the measurement range of the DSR was limited, so the initial test temperature of high dosage WEB-modified bitumen would be higher. For example, Figure 3a shows that even at 88 °C, the rutting factor of 10% WEB-modified bitumen was extremely large. If the starting temperature of 10% WEB-modified bitumen was set too low, the experimental results may have exceeded the DSR detection range. Another reason was also related to the dosage of the WEBs. Different WEB-modified bitumen had different failure temperatures. After failure, the viscosity of low dosage WEB-modified bitumen was too low for the DSR to detect and the DSR would stop the test and unload the sample automatically.

The rutting factor (G*/sinδ) is composed of complex modules and the phase angle. Therefore, the reason for the huge enhancement of the rutting factor should be analyzed from these two perspectives. The modification effect of the PE from WEB has been already demonstrated in other studies [21–25]. Generally speaking, the PE was harder than virgin bitumen. When it was blended with the virgin bitumen, it made the modified bitumen harder as the complex modules of the WEB-modified bitumen were enhanced. When the dosage of WEB reached 8%, the modifier was enough to cross-link with each other within the virgin bitumen to form a spatial network structure. When this structure was formed, the WEB-modified bitumen became less deformable under traffic load. The results were similar to the findings of Wang's study [23].

Additionally, the phase angle data of high dosage modified bitumen (over 8%) showed different trends. One distinction with normal bitumen was that the phase angle of the high dose WEB-modified bitumen decreased with the increase of temperature. As bitumen is viscous-elastic material, the phase angle will grow with the increasing temperature, regardless of whether it is virgin bitumen or modified bitumen. The viscous-elastic material's phase angle had the same tendency with temperature. This abnormal phenomenon may be due to two reasons. The modification mechanism of WEB-modified bitumen is that the matrix bitumen is filled with small particles of PE. First, before the temperature

reached 88 °C, the WEB-modified bitumen was blended in a stable status. When the temperature rose, the phase angle decreased. However, when the temperature reached 88 °C, the virgin bitumen began to melt, indicating that it could no longer resist the force. The DSR detector basically conducted the test on the PE portion covered by bitumen. The phase angle of the PE powder was significantly higher than the WEB-modified bitumen. This explains why the phase angle of the high-dosage WEB-modified bitumen decreased with increasing temperature at temperatures above 88 °C.

Similar to softening point the effect of $CaCO_3$ may also caused the improvement of high temperature performance. When the dosage of WEB reached a certain level, some virgin bitumen would combine with $CaCO_3$ and became modified asphalt mastic which may enhance the high temperature performance.

In addition to the Superpave rutting factor (G*/sinδ), the MSCR test was conducted for the rutting resistance ability of the WEB-modified bitumen. The test results are listed in Table 3. Three parameters were used for analysis including the average percent recovery (R), non-recoverable creep compliance ($J_{nr}$), and the stress sensitivity parameter ($J_{nr\text{-}diff}$). The percent recovery is an indicator of elastic behavior. Higher percent recovery indicates better elasticity. $J_{nr}$ presents the unrecoverable creep compliance of the bitumen binder at high temperature. A lower $J_{nr}$ indicates a superior capability to resist permanent deformation. The $J_{nr\text{-}diff}$ was used to assess the stress sensitivity of the bitumen binder. The MSCR results were in accordance with the $G^*/\sin\delta$ test result. According to Table 3, the WEB-modified bitumen showed a higher elastic behavior and lower unrecoverable creep compliance, which means it had better rutting resistance. According to the ASSHTO specification, bitumen with more than 6% of modifiers meets the AASHTO requirement for the highest traffic level "E", while bitumen with a 4% WEB meets the requirements of level "H". The virgin bitumen and modified bitumen with a 2% WEB only meets the qualifications of level "S".

**Table 3.** Multiple stress creep recovery (MSCR) test results with DSR at 64 °C.

| Binder Type | Temperature | $J_{nr0.1}$ | $J_{nr3.2}$ | $J_{nr\text{-}diff}$ | R | Traffic Level |
|:---:|:---:|:---:|:---:|:---:|:---:|:---:|
| Units | (°C) | (kPa$^{-1}$) | (kPa$^{-1}$) | (%) | (%) | / |
| Pen 60/70 | 58 | 3.977 | 4.768 | 19.9 | −4.4 | S |
| 2% WEB | 64 | 1.591 | 2.011 | 26.4 | −10.6 | S |
| 4% WEB | 64 | 0.692 | 1.083 | 56.5 | −0.4 | H |
| 6% WEB | 64 | 0.522 | 0.496 | −5.1 | −6.9 | E |
| 8% WEB | 64 | 0.027 | 0.164 | 500 | 36.4 | E |
| 10% WEB | 64 | 0.059 | 0.161 | 173.8 | 7.5 | E |

### 3.3. Low Temperature Performance

The stiffness value and the m-value are two key indicators to evaluate the low temperature performance obtained from the bending beam rheometer (BBR) test. Higher stiffness values mean more low-temperature cracking. According to the BBR result, the WEB additives had almost no negative effect on the low temperature performance. From some perspectives, it even improved the low temperature performance. In contrast to the SBS modified bitumen, the WEB-modified bitumen only had an effect on stiffness while the SBS modified bitumen could improve both the stiffness and m-value significantly [26]. Compared with PE, the WEB additives had a stronger improvement on stiffness, while the effect on the m-value was similar.

As shown in Table 4, the normal pen 60/70 bitumen met the AASHTO specifications until it reached −6 °C. At the same time, the WEB-modified bitumen also met the AASHTO specifications until it reached −6 °C. The incorporation of the WEB additives slightly decreased both the creep stiffness and m-value and they both tended to decrease with an increase in the dosage of the WEB modifier. As a consequence, the WEB-modified bitumen can be used more efficiently in regions where the lowest temperature is over 0 °C.

**Table 4.** Bending beam rheometer (BBR) test results.

| Binder Type | −6 °C | | −12 °C | |
|---|---|---|---|---|
| | Stiffness S (MPa) | m-Value | Stiffness S (MPa) | m-Value |
| Pen 60/70 | 191.15 | 0.355 | 323 | 0.257 |
| 2% WEB | 154.1 | 0.425 | 340.5 | 0.39 |
| 4% WEB | 141 | 0.309 | 275 | 0.264 |
| 6% WEB | 167 | 0.301 | 343.5 | 0.236 |
| 8% WEB | 154 | 0.325 | 248.5 | 0.250 |
| 10% WEB | 139.5 | 0.337 | 233 | 0.258 |

### 3.4. Workability

The workability of asphalt mixtures is a very broad concept concerning production, transport, laying, and compaction of asphalt mixtures. The workability of bitumen is generally evaluated by Rotational-viscosity (RV). Figure 4 presents the variation in the rotational viscosity of WEB-modified bitumen binders at different dosages and constant temperatures. This figure shows that the WEB modifier can increase the viscosity of the bitumen. Similar to the results of the softening point and rutting factors, the rotational viscosity increased steadily before 6% and showed great enhancement after 8%. The modified binder exhibited higher viscosity with increasing WEB dosage at 135 °C.

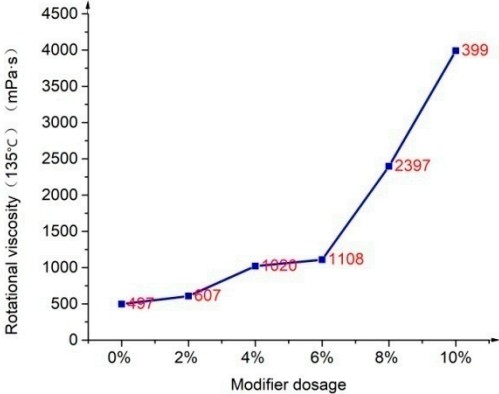

**Figure 4.** Rotational viscosity (135 °C).

### 3.5. Modification Mechanism Analysis

The modification mechanism of WEB-modified bitumen was further evaluated by Fourier transform infrared spectroscopy (FTIR). The FTIR test provides information on the chemical reaction during the modification.

When an organic compound molecule is irradiated with infrared light, a chemical bond or a functional group in the molecule can be absorbed by vibration. As different chemical bonds or functional groups absorb different frequencies, they will be in different positions in the infrared spectrum. Thus, information about the chemical bonds or functional groups contained in a molecule can be obtained. Figure 5 shows the FTIR spectrum of several typical samples in this research. Figure 5a indicates that WEB additives had two main absorption peaks at 2850 $cm^{-1}$ and 2921 $cm^{-1}$. Figure 5b shows that virgin bitumen had three main absorption peaks around 750 $^{-1}$, 1500 $^{-1}$, and 3000 $^{-1}$. However, when compared with Figure 5c, it can be seen that almost no obvious new absorption peaks were found in the 6% WEB-modified bitumen while the absorption peak around 1500 $cm^{-1}$ became much higher and wider. Also, in Figure 5d, no obvious new absorption peak appeared in 8% WEB-modified bitumen. And the absorption peak around 3500 $cm^{-1}$ became much higher and wider. This phenomenon may be caused by the high-speed shearing process, which led to an aging reaction and translated some aromatics and resins into asphaltenes [27,28]. No evident new absorption peak

indicated no obvious chemical reaction. Since no new harmful chemical substances were produced, the effect of WEB-modified bitumen on soil was similar to that of virgin bitumen, which means that the WEB-modified bitumen can be used at room temperature. Additionally, the WEB-modified bitumen was suitable to the mixture like SMA, which needs high module bitumen. These kinds of structures will protect the land by preventing the rain flow onto the land, therefore the roadbed soil will not be polluted by rainwater.

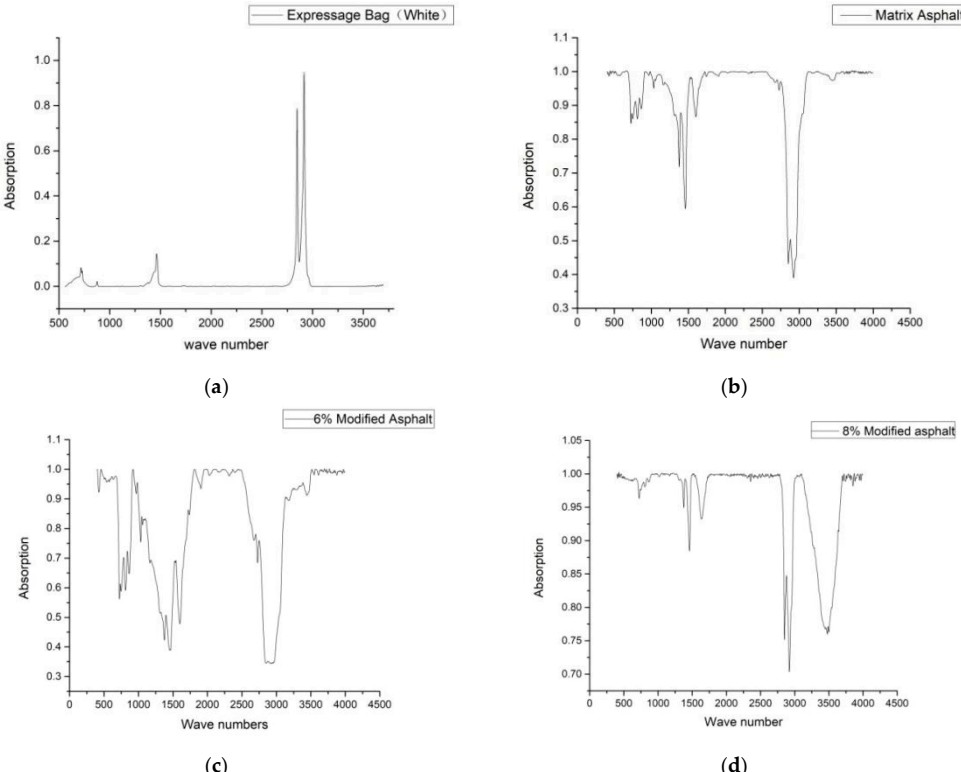

**Figure 5.** Results of the Fourier transform infrared spectroscopy (FTIR) tests: (**a**) Express plastic bags; (**b**) Matrix bitumen; (**c**) 6%WEB-modified bitumen; (**d**) 8%WEB-modified bitumen.

### 3.6. Morphology

The morphology of WEB-modified bitumen was evaluated by fluorescence microscope (FM) observations which providing information on the polymer in the modified bitumen. Both the bitumen and polymer will be lighter when they are exposed to fluorescence where the polymer will be brighter than the bitumen. Therefore, the distribution of the polymer can be observed. In the early study, it had been found out that after the modification, the SBS additives would become little particle and mixed with virgin bitumen homogeneously [29]. Interestingly, the FM experiment results of WEB-modified bitumen have similar conditions. Figure 6 shows the FM images of several typical samples in this research. It can be seen in Figure 6a that the virgin bitumen's image was very pure, indicating that the four fractions of virgin bitumen (saturates, aromatics, resins, and asphaltenes) were distributed homogeneously. The images of the 6% and 8% WEB-modified bitumen are shown in Figure 6b,c. In Figure 6b, many little polymer particles were homogeneously distributed in the bitumen. This demonstrates that the high-speed shearing process can make the WEB additives become much smaller and produce homogeneous WEB-modified bitumen. In Figure 6c, a new structure can be observed in the 8% WEB-modified bitumen. The WEB additives were combined together and became a bigger structure, which was like a skeleton structure. This "tiny aggregate" may significantly improve the rutting resistance of WEB-modified bitumen.

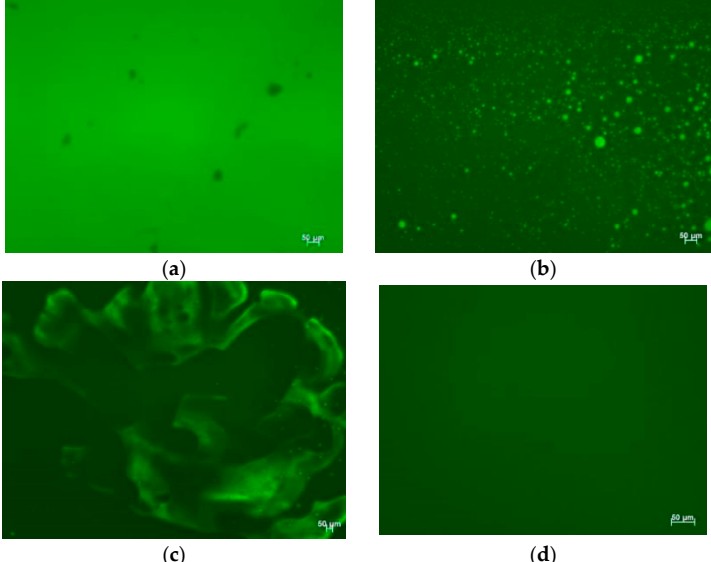

**Figure 6.** The fluorescence microscope (FM) images: (**a**) Pen 60/70; (**b**) modified bitumen with 6% WEB additive; (**c**) modified bitumen with 8% WEB additive; and (**d**) WEB additives.

## 4. Conclusions

This paper conducted a series of laboratory tests to characterize and compare the rheological properties of virgin bitumen modified with five different dosages of PE from waste express bags. Some materials experiments were also used to analyze the modification mechanism of the modified bitumen. According to the test data for the three empirical indicators, workability evaluation, Superpave rutting factor test, multiple stress creep recovery test, BBR test, Fourier transform infrared spectroscopy, and fluorescence microscope, the following major findings were obtained.

The softening point test, PG classification, and MSCR test revealed that WEB can greatly improve the high temperature performance of bitumen as a modifier. The incorporation of WEB additives improved the rutting resistance of virgin bitumen, and the rutting resistance increased with the increase in WEB dosage. This improvement comes from the modification effect by the waste express bags.

When the percentage of the express plastic bag modifier reached a certain level ($\geq 8\%$), the high temperature performance of WEB-modified bitumen was significantly improved. One reason for this is that the PE particles combined to become a bigger structure. The second reason is that the phase angle of the bitumen decreased with the temperature increase. This may be due to the effect of the PE skeleton, which plays an important role in high temperature environments.

The WEB modifier showed a negative effect on ductility while the data from the BBR test showed that the low temperature performance of WEB-modified bitumen did not decrease when compared with pen 60/70. Therefore, the WEB modifier is more applicable in regions where the lowest temperature is over 0 °C.

The WEB additives may decrease the workability to some extent. Therefore, WEB-modified bitumen can be used when the WEB dosage is under 8%.

The Fourier transform infrared spectroscopy tests showed that the waste express bags did not have a significant chemical reaction during the process of modification. The modification mechanism is mainly physical blending, which means that the WEB-modified bitumen's influence on soil is similar to that of virgin bitumen. It is safe at room temperature.

Fluorescence microscopy showed that at low WEB dosages, the PE will mix with the virgin bitumen in the form of little particles. At high WEB dosages, the PE will combine together to become a skeleton structure. This may explain why the high temperature performance increased significantly when the additives reached a high level.

On account of the limited findings of this research, the use of waste express bags as a bitumen additive appears promising. WEB additives have been found to possibly improve the performance of asphalt mixtures in both rutting resistance and low temperature resistance. As plenty of road construction is being planned in China, the use of WEB as a bitumen modifier will surely help with waste management. Further research is needed on WEB-modified asphalt mixture experiments and field performance evaluation. Additionally, many other observation techniques can be used to survey the internal morphology of the modified binders in further study. Finally, WEB-modified asphalt toxicity analysis, recyclability research of WEB and life cost assessment (LCA) study of WEB-modified asphalt pavement can be conducted in the further research.

**Author Contributions:** Experimental design, C.H.; performing of experiments, Y.L.; formal analysis, Y.L., S.A., C.H and C.W.; writing of paper, Y.L., C.H., M.Y and S.A.

**Funding:** This research was funded by the National Natural Science Foundation [51578248] and Pearl River S&T Nova Program of Guangzhou [201710010063].

**Acknowledgments:** Support provided by the professor Zhanping You was greatly appreciated.

**Conflicts of Interest:** The authors declare no conflicts of interest.

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
