# Peer review of "Evaluation of Waste Express Bag as a Novel Bitumen Modifier"

_applsci, doi:10.3390/app9061242_

Round 1

Reviewer 1 Report

Review of paper entitled "Evaluation of Waste Express Bag as a Novel Bitumen Modifier" authored by Lin et al. (2019)

The paper focuses on evaluating the feasibility of using WEB as novel bitumen modifier for hot mix asphalt. Various experimental analyses including penetration, softening point and ductility tests were conducted to probe the physical properties of different dosages of WEB modified asphalt. Additionally,the rheological properties of different samples were tested. 

Overall, this is an interesting paper and can be published in this journal. However, the authors should carefully address my comments/suggestions as reported below:

(1) With regards to the modifier dose, why did the authors only study the range between 0-10%? Although authors stated it had poor performance, It is still important that the authors add more details, accordingly. 

(2) It is interesting that as modifier dose increases from ~6 to 10%, the softening point increases to a large extent (i.e. from ~59 to 90 C. Why is it the case? Is it just because of cross-linking happening or some new complex structures/species (very large form of aggregates/precipitates) are forming? Please provide more discussions in this regard.

(3) What is the rheological nature of these bitumen samples? Are they newtonian, non-Newtonian? 

(4) The error of viscosity measurements should be stated or added into the main text. 

(5) The experimental conditions need to be added into the caption of Figure 4.

(6)  Please improve the quality of all figures in the paper (for example Figure 2). If possible, it would be nice to use different colors (similar to Figure 3) for the data points and larger fonts. 

(7) The manuscript needs to be proofread. Few grammatical issues as well as typos can be found in the manuscript. 

(8) The literature review should be improved in the manuscript. The following research articles need to be cited at least in the paper: 

(i) Energy & Fuels, 29(9), pp.5595-5599. (ii) Colloids and Surfaces A: Physicochemical and Engineering Aspects, 513, pp.178-187. (iii) Physical Review E, 96(5), p.052803 (iv) Langmuir 33.8 (2017): 1927-1942.

Author Response

Great thanks for your valuable comments.

Please check the following response.

Thanks a lot.

                                                                                                                                           Chichun Hu

                                                                                                                                        11st Mar 2019

Reviewer 2 Report

Dear authors

thank you for your interesting paper on the use of Waste Express Bags as bitumen modifier.

In the uploaded file you can find all my minor comments/questions and markings related to the English quality of the paper (which should be revised by a native speaker or specialized language editing service).

Here I will summarize my main comments (not related to language):

1) Introduction: can you include some more details, if available, on previous studies where LDPE was used as modifier, including the rheological results (or compare to them when discussing your own results)

2) Table 1: add the references to the standards in this table (like you did in Table 2) and remove all references to standards from the reference section + are these your own test results or taken from the supplier?

3) Rolling viscosity: is this the right term? not rotational/rotating viscosity? Similar remark related to the use of binder/bitumen/asphalt ==> make sure you use the right terminology.

4) Figure 2: you mixed for 1 hour at 180°C ==> what is the influence of this mixing process on the virgin binder itself? This would be interesting to check and add to the results.

5) effect of CaCO3: you claim this as a possible explanation, but what did you find in literature on this topic?

6) Figure 3: phase angle is unclear; especially why you have only some results at certain temperatures + the significance/interpretation of these results should be elaborated? Also, add some actual numbers on the Y-axis and compare these results with literature.

7) Lay-out of Table 3 should be improved + add some statistical analysis (if possible, e.g. multiple samples ==> standard deviation should be included)

8) Table 4: also here a discussion should include more than summarizing the results or to state that a AASHTO specification has been met - please compare with results from literature, e.g. compare the results with a SBS modified PmB or preferably  with LDPE modified bitumen

9) FTIR results: very difficult to quantitatively compare FTIR results, so this should be improved/investigated further. I also do not understand how you can claim that no harmful chemical substances are present by analysing FTIR results. When heating VOC's should be investigated and leaching tests should be performed to make sure no harmful substances leach in the soil? Fig. 5 should be improved / expanded as well

10) Fluorescence microscopy: nice pictures, but I think "tiny aggregate" is not the right teminology + room to add a fourth picture. Also here, compare with results from literature where FM has been used as well on modified binders.

11) Conclusions should be reformulated when more comparisons with literature were made + statements related to environmental issues should be reformulated. As mentioned before, I believe more tests (VOC/leaching) are needed and a suggestion for further research could include an LCA-study to investigate its overal environmental impact, compared to virgin or PmB binders. Finally, unclear how foaming or WMA-additives will improve the workability

12) References: please review the way you refer to the references and check that you actually refer to all of them in the text. I am quite sure that the last references are not used in the text at the moment. There is also one reference included twice and the standards should be removed from the list.

Good luck!

Author Response

(The authors gave the same response as above.)

Round 2

Reviewer 1 Report

I would like to thank the authors for addressing the majority of my comments. While I recommend the paper to be published in its present format, I recommend authors to do the final minor comments that can be done in a timely fashion: 

(i) Make figure 5  larger as it is difficult to see the peaks. 

(ii) Please cite few articles published in Applied Sciences, and (iii) Check the references again and also cite the following two papers to acknowledge the contribution of these groups to the field: 1) Energy & Fuels, 29(9), pp.5595-5599. 2) Langmuir 33.8 (2017): 1927-1942

Author Response

Great thanks for your valuable comments.

Please check the following response.

Thanks a lot.

                                                                                                                                           Chichun Hu

                                                                                                                                       15th Mar 2019

Reviewer 2 Report

Dear authors

thank you for the revised version. I am pleased with most of the corrections, but have a few comments remaining:

- Line 83: asphalt concrete, not bitumen concrete! Check if asphalt and bitumen are used correctly throughout the text. I think you did a find and replace, but it should be decided case by case.

- Lines 80-82: check spacing, some words are "glued" together: carbonatecould and carbonateis (similar remark for line 123, so check the whole text for these kind of errors)

- Line 112: some text below the table + normally table subscript is above the table?

- Table 3: header - I would propose to use the fact that you have two lines for the header, so that you place the units on the second line + why put Temperature which has to be split then ==> use T and also unit °C on second line + are 5 decimals really worth mentioning/relevant?

- Figure 5: in my comments in the pdf, I suggested to fill that empty space by an additional subfigure (d), e.g. FTIR for 2% or 10% (only, if available!)

- Conclusions: WMA and foam bitumen reduce temperature during construction, but that should have no impact on the performance of the asphalt. Performance and workability are also two different things! Workability is related to how easily it can be compacted. I suggest to remove this sentence from the conclusions as I don't see its relevance.

- Lines 279-284 Toxicity of WEB: I still don’t see the link between an FTIR-test at room temperature, and possible fumes appearing when heating the bitumen and asphalt above 160 °C, or environmental issues after installation or during recycling. I still believe it is a dangerous statement. I am concerned about the safety of the asphalt workers (fumes) and the environment (leaching / recycling issues). We don't want additional small plastic particles floating in our environment, or be transported in our water, when small particles would be removed through the effect of the traffic on the road. Additionally, we don't want another "tar" situation, where this material cannot be recycled any more. What if this is also true for WEB modified asphalt?

I recommend to add some comments on these matters for future research in the conclusions (LCA study, further analysis of toxicity and H&S issues, study of the recyclability).

I accept the paper in the current form, but I hope you take the comments above and deal with them in the final version (including a thorough language revision).

good luck!

Author Response

(The authors gave the same response as above.)
